# *ColPali*: Efficient Document Retrieval with Vision Language Models

**Manuel Faysse**[*1,3]   **Hugues Sibille**[*1,4]   **Tony Wu**[*1]   **Bilel Omrani**[1]
**Gautier Viaud**[1]   **Céline Hudelot**[3]   **Pierre Colombo**[2,3]
[1]Illuin Technology   [2]Equall.ai   [3]CentraleSupélec, Paris-Saclay   [4]ETH Zürich
manuel.faysse@centralesupelec.fr

## Abstract

Documents are visually rich structures that convey information through text, but also figures, page layouts, tables, or even fonts. Since modern retrieval systems mainly rely on the textual information they extract from document pages to index documents -often through lengthy and brittle processes-, they struggle to exploit key visual cues efficiently. This limits their capabilities in many practical document retrieval applications such as Retrieval Augmented Generation (RAG). To benchmark current systems on visually rich document retrieval, we introduce the Visual Document Retrieval Benchmark *ViDoRe*, composed of various page-level retrieval tasks spanning multiple domains, languages, and practical settings. The inherent complexity and performance shortcomings of modern systems motivate a new concept; doing document retrieval by directly embedding the images of the document pages. We release *ColPali*, a Vision Language Model trained to produce high-quality multi-vector embeddings from images of document pages. Combined with a late interaction matching mechanism, *ColPali* largely outperforms modern document retrieval pipelines while being drastically simpler, faster and end-to-end trainable. We release models, data, code and benchmarks under open licenses at https://hf.co/vidore.

## 1 Introduction

Document Retrieval consists of matching a user query to relevant documents in a given corpus. It is central to many widespread industrial applications, either as a standalone ranking system (search engines) or as part of more complex information extraction or Retrieval Augmented Generation (RAG) pipelines.

Over recent years, pretrained language models have enabled large improvements in text embedding models. In practical industrial settings, however, the primary performance bottleneck for efficient document retrieval stems not from embedding model performance but from the prior data ingestion pipeline. Indexing a standard PDF document involves several steps. First, PDF parsers or Optical Character Recognition (OCR) systems are used to extract words from the pages. Document layout detection models can then be run to segment paragraphs, titles, and other page objects such as tables, figures, and headers. A chunking strategy is then defined to group text passages with some semantical coherence, and modern retrieval setups may even integrate a captioning step to describe visually rich elements in a natural language form, more suitable for embedding models. In our experiments (Table 2), we typically find that optimizing the ingestion pipeline yields much better performance on visually rich document retrieval than optimizing the text embedding model.

**Contribution 1: *ViDoRe*.** In this work, we argue that document retrieval systems should not be evaluated solely on the capabilities of text embedding models (Bajaj et al., 2016; Thakur et al., 2021; Muennighoff et al., 2022), but should also consider the context and visual elements of the documents to be retrieved. To this end, we create and openly release *ViDoRe*, a comprehensive benchmark to evaluate systems on page-level document retrieval with a wide coverage of domains, visual elements, and languages. *ViDoRe* addresses practical document retrieval scenarios, where

---

[*]Equal Contribution

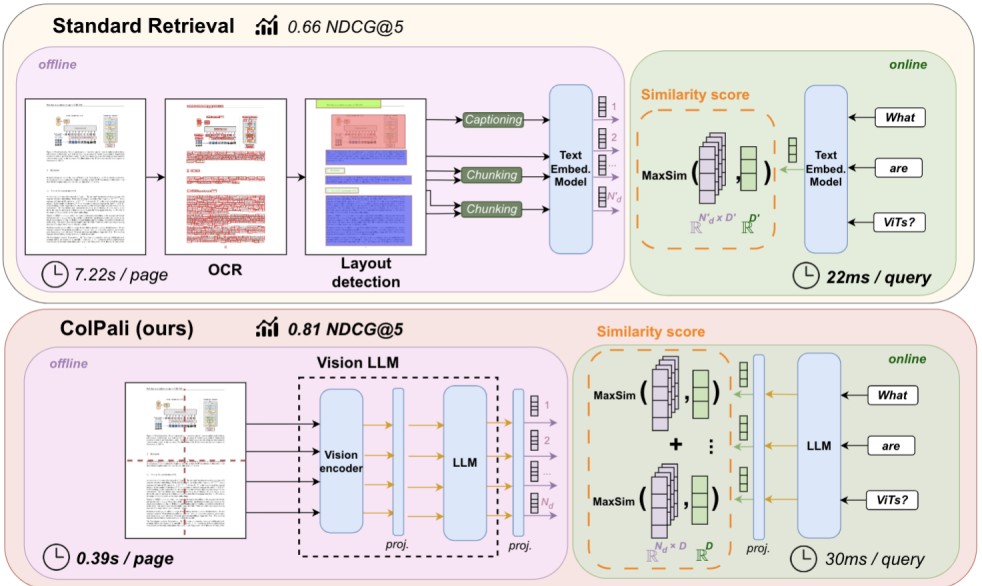

Figure 1: *ColPali* simplifies document retrieval w.r.t. standard retrieval methods while achieving stronger performances with better latencies. Latencies and results are detailed in section 5 and subsection B.4.

queries often necessitate both textual and visual understanding for accurate document matching. We highlight the shortcomings of current text-centric systems in these settings.[1]

**Contribution 2:** *ColPali*. We propose a novel concept and model architecture based on Vision Language Models (VLMs) to efficiently index documents purely from their visual features, allowing for subsequent fast query matching with late interaction mechanisms (Khattab & Zaharia, 2020). Our method, *ColPali*, significantly outperforms all other retrieval systems on *ViDoRe* while being fast and end-to-end trainable. These results demonstrate the potential and the many benefits of this novel *Retrieval in Vision Space* concept, which could significantly alter the way document retrieval is approached in the industry moving forward. We release all resources at `https://hf.co/vidore`.

## 2 PROBLEM FORMULATION & RELATED WORK

**Problem Setting.** In our setting, a retrieval system scores how relevant a document $d$ from corpus $\mathcal{D}$ is with respect to a query $q$. Computing the similarity score $s(q, d) \in \mathbb{R}$ for each of the $|\mathcal{D}|$ documents in the corpus creates a ranking we can use to extract the most relevant documents. In this work, we focus on page-level retrieval: *given a query, is the correct document page retrieved by the system?* For coherence with existing literature, we further use the term *document* to refer to individual pages, i.e. the atomic retrieved elements in our setting. As we focus on practical industrial retrieval applications (RAG, search engines) with potentially large corpora sizes, latency constraints are imposed on scoring systems. Most current retrieval systems can be decomposed into (1) an *offline* indexation phase in which a document index is built and (2) an *online* querying phase in which a query is matched to documents from the index and where low latency is vital to the user experience.

*Under these industrial constraints, we identify three main properties an efficient document retrieval systems should exhibit:* ***(R1) strong retrieval performance****, as measured by standard retrieval metrics;* ***(R2) fast online querying****, measured through average latencies;* ***(R3) high throughput corpus indexation****, ie. the number of pages that can be embedded in a given timeframe.*

---

[1] The *ViDoRe* benchmark leaderboard is hosted publicly at `https://huggingface.co/spaces/vidore/vidore-leaderboard` to encourage further developments.

## 2.1 TEXTUAL RETRIEVAL METHODS

**Document Retrieval in Text Space.**

Statistical methods based on word frequency like TF-IDF (Sparck Jones, 1972) and BM25 (Robertson et al., 1994) are still widely used due to their simplicity and efficiency. More recently, neural embedding models based on fine-tuned large language models display state-of-the-art performance on a variety of text embedding tasks and top the retrieval leaderboards (Muennighoff et al., 2022).

**Neural Retrievers.** In bi-encoder models (Reimers & Gurevych, 2019; Karpukhin et al., 2020; Wang et al., 2022), documents are independently mapped *offline* to a dense vector space. Queries are embedded *online* and matched to documents through a fast cosine distance computation. A slower, but slightly more performant alternative, cross-encoder systems (Wang et al., 2020; Cohere, 2024) concatenate query and document as a single input sequence and iteratively attribute matching scores to each possible combination. This enables full attention computation between query and document terms but comes at the cost of computational efficiency, as $|\mathcal{D}|$ encoding passes must be done online.

**Multi-Vector retrieval via late interaction.** In the late interaction paradigm introduced by ColBERT (Khattab & Zaharia, 2020), an embedding is pre-computed and indexed per document token. At runtime, similarity can be computed with individual query token embeddings. The idea is to benefit from the rich interaction between individual query and document terms while taking advantage of the offline computation and fast query matching enabled by bi-encoders. See section E for more details.

**Retrieval Evaluation.** Although benchmarks and leaderboards have been developed to evaluate text embedding models (Thakur et al., 2021; Muennighoff et al., 2022), much of the performance improvements in industrial use cases of embedding models stem from the prior data ingestion pipeline. While documents often rely on visual elements to more efficiently convey information to human readers, text-only systems barely tap into these visual cues. Other work has also independently studied table or chart retrieval systems through repurposed Question Answering datasets (Zhang et al., 2019; Nowak et al., 2024) but only assessing specialized methods for each task.

*To our knowledge, no benchmark evaluates document retrieval systems in practical settings; in an end-to-end manner, across several document types and topics, and by evaluating the use of both textual and visual document features.*

## 2.2 INTEGRATING VISUAL FEATURES

**Contrastive Vision Language Models.** Mapping latent representations of textual content to corresponding representations of visual content has been done by aligning disjoint visual and text encoders through contrastive losses (Radford et al., 2021; Zhai et al., 2023). While some OCR capabilities exist in these models, the visual component is often not optimized for text understanding.

The Fine-grained Interactive Language-Image Pre-training (Yao et al., 2021) framework extends the late interaction mechanism to cross-modal Vision Language Models, relying on max similarity operations between text tokens and image patches.

**Visually Rich Document Understanding.** To go beyond text, some document-focused models jointly encode text tokens alongside visual or document layout features (Appalaraju et al., 2021; Kim et al., 2021; Huang et al., 2022; Tang et al., 2022). Large Language transformer Models (LLMs) with strong reasoning capabilities have recently been combined with Vision Transformers (ViTs) (Dosovitskiy et al., 2020) to create VLMs (Alayrac et al., 2022; Liu et al., 2023; Bai et al., 2023; Laurençon et al., 2024b) where image patch vectors from contrastively trained ViT models (Zhai et al., 2023) are fed as input embeddings to the LLM and concatenated with the text-token embeddings.

**PaliGemma.** The PaliGemma-3B model (Beyer et al., 2024) extends concepts from Pali3 (Chen et al., 2023), and projects `SigLIP-So400m/14` (Alabdulmohsin et al., 2023) patch embeddings into Gemma-2B's text vector space (Gemma Team et al., 2024). Along with its reasonable size w.r.t. other performant VLMs, an interesting property of PaliGemma's text model is that it is fine-tuned with full-block attention on the prefix (instruction text and image tokens). See Appendix E for more details.

*VLMs display enhanced capabilities in Visual Question Answering, captioning, and document understanding (Yue et al., 2023), but are not optimized for retrieval tasks.*

## 3 THE *ViDoRe* BENCHMARK

Existing benchmarks for contrastive vision-language models primarily evaluate retrieval for natural images (Lin et al., 2014; Borchmann et al., 2021; Thapliyal et al., 2022). On the other hand, textual retrieval benchmarks (Muennighoff et al., 2022) are evaluated at at textual passage level and are not tailored for document retrieval tasks. We fill the gap with *ViDoRe*, a comprehensive benchmark for document retrieval using visual features.

### 3.1 BENCHMARK DESIGN

*ViDoRe* is designed to comprehensively evaluate retrieval systems on their capacity to match queries to relevant documents at the page level. This benchmark encompasses multiple orthogonal subtasks, with focuses on various modalities - text, figures, infographics, tables; thematic domains - medical, business, scientific, administrative; or languages - English, French. Tasks also span varying levels of complexity, in order to capture signals from both weaker and stronger systems. As many systems require large amounts of time to index pages (captioning-based approaches can take dozens of seconds per page for instance), we limit the number of candidate documents for each retrieval task in order to evaluate even complex systems in a reasonable timeframe without sacrificing quality. For trainable retrieval systems, we provide a reference training set that can be used to facilitate comparisons.

| Dataset | Language | # Queries | # Documents | Description |
|---------|----------|-----------|-------------|-------------|
| **Academic Tasks** | | | | |
| DocVQA | English | 500 | 500 | Scanned documents from UCSF Industry |
| InfoVQA | English | 500 | 500 | Infographics scrapped from the web |
| TAT-DQA | English | 1600 | 1600 | High-quality financial reports |
| arXiVQA | English | 500 | 500 | Scientific Figures from arXiv |
| TabFQuAD | French | 210 | 210 | Tables scrapped from the web |
| **Practical Tasks** | | | | |
| Energy | English | 100 | 1000 | Documents about energy |
| Government | English | 100 | 1000 | Administrative documents |
| Healthcare | English | 100 | 1000 | Medical documents |
| AI | English | 100 | 1000 | Scientific documents related to AI |
| Shift Project | French | 100 | 1000 | Environmental reports |

Table 1: *ViDoRe* comprehensively evaluates multimodal retrieval methods.

**Academic Tasks.** We repurpose widely used visual question-answering benchmarks for retrieval tasks: for each page-question-answer triplet, we use the question as the query, and the associated page as the gold document (Table 1). These academic datasets either focus on single specific modalities (Mathew et al., 2020; 2021; Li et al., 2024) or target more varied visually rich documents (Zhu et al., 2022). Moreover, we consider TabFQuAD, a human-labeled dataset on tables extracted from French industrial PDF documents released with this work. Details can be found in subsection A.1.

**Practical tasks.** We construct topic-specific retrieval benchmarks spanning multiple domains to go beyond repurposed QA datasets and evaluate retrieval in more realistic industrial situations (e.g. RAG). To achieve this, we collect publicly accessible PDF documents and generate queries pertaining to document pages using Claude-3 Sonnet, a high-quality proprietary vision-language model (Anthropic, 2024). In total, we collect 1,000 document pages per topic, which we associate with 100 queries extensively filtered for quality and relevance by human annotators. The corpus topics are intentionally specific to maximize syntactic proximity between documents, creating more challenging retrieval tasks and covering an array of orthogonal domains (Table 1). [2]

**Evaluation Metrics.** We evaluate performance on our benchmark (Requirement R1) using standard metrics from the retrieval literature (nDCG, Recall@K, MRR). We report nDCG@5 values as the main performance metric in this work and release the complete sets of results along with the models[3].

---

[2] Answers are generated alongside queries to (1) ground queries and improve their quality and (2) provide resources to foster future work.

[3] https://huggingface.co/vidore

To validate compliance with practical industrial requirements (section 2), we also consider query latencies (R2) and indexing throughputs (R3).

## 3.2 ASSESSING CURRENT SYSTEMS

**Unstructured.** We evaluate retrieval systems representative of those found in standard industrial RAG pipelines. As is common practice, we rely on the `Unstructured`[4] off-the-shelf tool in the highest resolution settings to construct high-quality text chunks from PDF documents. `Unstructured` orchestrates the document parsing pipeline, relying on deep learning vision models to detect titles and document layouts (Ge et al., 2021), OCR engines (Smith, 2007) to extract text in non-native PDFs, specialized methods or models to detect and reconstruct tables, and implements a chunking strategy (`by-title`) that leverages the detected document structure to preserve section boundaries when concatenating texts. As is common practice, in our simplest `Unstructured` configuration (*text-only*), only textual elements are kept and figures, images, and tables are considered noisy information and are filtered out.

**Unstructured + X.** While `Unstructured` is a strong baseline by itself, we further augment `Unstructured`'s output by integrating the visual elements. In (*+ OCR*), tables, charts, and images are run through an OCR engine, processed by Unstructured, and chunked independently. In (*+ Captioning*), we set up a fully-fledged captioning strategy (Zhao et al., 2023), in which we feed visual elements to a strong proprietary Vision Language Model (Claude-3 Sonnet (Anthropic, 2024)) to obtain highly detailed textual descriptions of the elements. Both strategies aim to integrate visual elements in the retrieval pipeline but incur significant latency and resource costs (subsection 5.2).

**Embedding Model.** To embed textual chunks, we evaluate Okapi BM25, the *de facto* standard sparse statistical retrieval method, and the dense encoder of BGE-M3 (Chen et al., 2024), a multilingual neural method with SOTA performance in its size category. Chunks are embedded and scored independently, and page-level scores are obtained by max-pooling over the page's chunk scores.[5]

**Contrastive VLMs.** We also evaluate the strongest available vision-language embedding models; Jina CLIP (Koukounas et al., 2024), Nomic Embed Vision (Nomic, 2024), and `SigLIP-So400m/14` (Alabdulmohsin et al., 2023).

**Results.** From a performance perspective, best results are obtained by combining the `Unstructured` parser with visual information, either from captioning strategies or by running OCR on the visual elements (Table 2). Little difference is seen between BM25 and BGE-M3 embeddings highlighting the visual information bottleneck. Contrastive VLMs lag behind. Beyond retrieval performance (R1), the indexing latencies (R2) reported in Figure 2 illustrate that PDF parsing pipelines can be very lengthy, especially when incorporating OCR or captioning strategies. Querying latencies at runtime (R3) are very good for all evaluated systems ($\leq$ 22 ms on a NVIDIA L4) due to fast query encoding and cosine similarity matching.

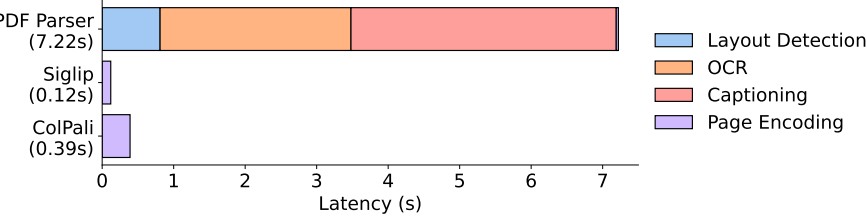

Figure 2: Offline document indexing with *ColPali* is much simpler and faster compared to standard retrieval methods. The PDF Parser results are obtained following the *Unstructured* settings with BGE-M3 detailed in subsection 3.2. All indexing speeds are averaged per-page latencies. More details in subsection B.4

.

---

[4]`www.unstructured.io`

[5]We empirically validated the max-pooling strategy over sub-page chunks to be more effective than concatenating all page chunks before embedding pagewise.

## 4 LATE INTERACTION BASED VISION RETRIEVAL

### 4.1 ARCHITECTURE

**Vision-Language Models.** Encouraged by their strong document understanding capabilities, we propose adapting recent VLMs for retrieval. The key concept is to leverage the alignment between output embeddings of text and image tokens acquired during multi-modal fine-tuning. To this extent, we introduce *ColPali*, a Paligemma-3B extension that is capable of generating ColBERT-style multi-vector representations of text and images (Figure 1). PaliGemma-3B is a strong candidate due to its small size, the many released checkpoints fine-tuned for different image resolutions and tasks, and the promising performances on various document understanding benchmarks. We add a projection layer to map each of the language model's output token embeddings (whether from text or image tokens) to a vector space of reduced dimension $D = 128$ as used in the ColBERT paper (Khattab & Zaharia, 2020) to keep lightweight bag-of-embedding representations.

**Late Interaction.** Given query $q$ and document $d$, we denote as $\mathbf{E_q} \in \mathbb{R}^{N_q \times D}$ and $\mathbf{E_d} \in \mathbb{R}^{N_d \times D}$ their respective multi-vector representation in the common embedding space $\mathbb{R}^D$, where $N_q$ and $N_d$ are respectively the number of vectors in the query and in the document page embeddings. The late interaction operator, $\text{LI}(q, d)$, is the sum over all query vectors $\mathbf{E_q}^{(j)}$, of its maximum dot product $\langle \cdot | \cdot \rangle$ with each of the $N_d$ document embedding vectors $\mathbf{E_d}_{(1:N_d)}$.

$$\text{LI}(q, d) = \sum_{i \in [|1, N_q|]} \max_{j \in [|1, N_d|]} \langle \mathbf{E_q}^{(i)} | \mathbf{E_d}^{(j)} \rangle \tag{1}$$

**Contrastive Loss.** The Late Interaction operation is fully differentiable, enabling backpropagation. Let a batch $\{q_k, d_k\}_{k \in [|1, b|]}$ composed of $b$ query-page pairs, where for all $k \in [|1, b|]$, the document page $d_k$ is the document corresponding to query $q_k$. Following Khattab & Zaharia (2020), we define our in-batch contrastive loss $\mathcal{L}$ as the softmaxed cross-entropy of the positive scores $s_k^+ = \text{LI}(q_k, d_k)$ w.r.t. to the maximal in-batch negative scores $s_k^- = \max_{l, l \neq k} \text{LI}(q_k, d_l)$[6]:

$$\mathcal{L} = -\frac{1}{b} \sum_{k=1}^{b} \log \left[ \frac{\exp(s_k^+)}{\exp(s_k^+) + \exp(s_k^-)} \right] = \frac{1}{b} \sum_{k=1}^{b} \log \left( 1 + \exp(s_k^- - s_k^+) \right) \tag{2}$$

### 4.2 MODEL TRAINING

**Dataset.** Our training dataset of 118,695 query-page pairs is comprised of train sets of openly available academic datasets (63%) and a synthetic dataset made up of pages from web-crawled PDF documents and augmented with VLM-generated (Claude-3 Sonnet) pseudo-questions (37%). Dataset split details are given in subsection A.3. Our training set is fully English by design, enabling us to study zero-shot generalization to non-English languages[7]. We explicitly verify no multi-page PDF document is used both *ViDoRe* and in the train set to prevent evaluation contamination. A validation set is created with 2% of the samples to tune hyperparameters. We openly release the training dataset[8] for reproducibility and to encourage further research.

**Parameters.** All models are trained for 1 epoch on the train set. Unless specified otherwise, we train models in `bfloat16` format, use low-rank adapters (LoRA, Hu et al. (2021)) with $\alpha = 32$ and $r = 32$ on the transformer layers from the language model, as well as the final randomly initialized projection layer, and use a `paged_adamw_8bit` optimizer. We train on an 8 GPU setup with data parallelism, a learning rate of $5e - 5$ with linear decay with 2.5% warmup steps, and a batch size of 32.

**Query Augmentation.** As in Khattab & Zaharia (2020), we append 5 `<unused0>` tokens to the query tokens to serve as a soft, differentiable query expansion or re-weighting mechanism.

---

[6]We reformulate the loss to leverage the numerically stable `softplus` function where `softplus(x)` $= \log(1 + \exp(x))$

[7]Multilingual data is present in the pretraining corpus of the language model (Gemma-2B) and potentially occurs during PaliGemma-3B's multimodal training.

[8]https://huggingface.co/datasets/vidore/colpali_train_set

# 5 RESULTS

|  | ArxivQ | DocQ | InfoQ | TabF | TATQ | Shift | AI | Energy | Gov. | Health. | Avg. |
|---|---|---|---|---|---|---|---|---|---|---|---|
| **Unstructured** text-only |  |  |  |  |  |  |  |  |  |  |  |
| - BM25 | - | 34.1 | - | - | 44.0 | 59.6 | 90.4 | 78.3 | 78.8 | 82.6 | - |
| - BGE-M3 | - | 28.4$_{\downarrow5.7}$ | - | - | 36.1$_{\downarrow7.9}$ | 68.5$_{\uparrow8.9}$ | 88.4$_{\downarrow2.0}$ | 76.8$_{\downarrow1.5}$ | 77.7$_{\downarrow1.1}$ | 84.6$_{\uparrow2.0}$ | - |
| **Unstructured** + OCR |  |  |  |  |  |  |  |  |  |  |  |
| - BM25 | 31.6 | 36.8 | 62.9 | 46.5 | 62.7 | 64.3 | 92.8 | 85.9 | 83.9 | 87.2 | 65.5 |
| - BGE-M3 | 31.4$_{\downarrow0.2}$ | 25.7$_{\downarrow11.1}$ | 60.1$_{\downarrow2.8}$ | 70.8$_{\uparrow24.3}$ | 50.5$_{\downarrow12.2}$ | 73.2$_{\uparrow8.9}$ | 90.2$_{\downarrow2.6}$ | 83.6$_{\downarrow2.3}$ | 84.9$_{\uparrow1.0}$ | 91.1$_{\uparrow3.9}$ | 66.1$_{\uparrow0.6}$ |
| **Unstructured** + Captioning |  |  |  |  |  |  |  |  |  |  |  |
| - BM25 | 40.1 | 38.4 | 70.0 | 35.4 | 61.5 | 60.9 | 88.0 | 84.7 | 82.7 | 89.2 | 65.1 |
| - BGE-M3 | 35.7$_{\downarrow4.4}$ | 32.9$_{\downarrow5.4}$ | 71.9$_{\uparrow1.9}$ | 69.1$_{\uparrow33.7}$ | 43.8$_{\downarrow17.7}$ | 73.1$_{\uparrow12.2}$ | 88.8$_{\uparrow0.8}$ | 83.3$_{\downarrow1.4}$ | 80.4$_{\downarrow2.3}$ | 91.3$_{\uparrow2.1}$ | 67.0$_{\uparrow1.9}$ |
| **Contrastive VLMs** |  |  |  |  |  |  |  |  |  |  |  |
| Jina-CLIP | 25.4 | 11.9 | 35.5 | 20.2 | 3.3 | 3.8 | 15.2 | 19.7 | 21.4 | 20.8 | 17.7 |
| Nomic-vision | 17.1 | 10.7 | 30.1 | 16.3 | 2.7 | 1.1 | 12.9 | 10.9 | 11.4 | 15.7 | 12.9 |
| SigLIP (Vanilla) | 43.2 | 30.3 | 64.1 | 58.1 | 26.2 | 18.7 | 62.5 | 65.7 | 66.1 | 79.1 | 51.4 |
| **Ours** |  |  |  |  |  |  |  |  |  |  |  |
| SigLIP (Vanilla) | 43.2 | 30.3 | 64.1 | 58.1 | 26.2 | 18.7 | 62.5 | 65.7 | 66.1 | 79.1 | 51.4 |
| BiSigLIP (+fine-tuning) | 58.5$_{\uparrow15.3}$ | 32.9$_{\uparrow2.6}$ | 70.5$_{\uparrow6.4}$ | 62.7$_{\uparrow4.6}$ | 30.5$_{\uparrow4.3}$ | 26.5$_{\uparrow7.8}$ | 74.3$_{\uparrow11.8}$ | 73.7$_{\uparrow8.0}$ | 74.2$_{\uparrow8.1}$ | 82.3$_{\uparrow3.2}$ | 58.6$_{\uparrow7.2}$ |
| BiPali (+LLM) | 56.5$_{\downarrow2.0}$ | 30.0$_{\downarrow2.9}$ | 67.4$_{\downarrow3.1}$ | 76.9$_{\uparrow14.2}$ | 33.4$_{\uparrow2.9}$ | 43.7$_{\uparrow17.2}$ | 71.2$_{\downarrow3.1}$ | 61.9$_{\downarrow11.7}$ | 73.8$_{\downarrow0.4}$ | 73.6$_{\downarrow8.8}$ | 58.8$_{\uparrow0.2}$ |
| *ColPali* (+Late Inter.) | **79.1**$_{\uparrow22.6}$ | **54.4**$_{\uparrow24.5}$ | **81.8**$_{\uparrow14.4}$ | **83.9**$_{\uparrow7.0}$ | **65.8**$_{\uparrow32.4}$ | **73.2**$_{\uparrow29.5}$ | **96.2**$_{\uparrow25.0}$ | **91.0**$_{\uparrow29.1}$ | **92.7**$_{\uparrow18.9}$ | **94.4**$_{\uparrow20.8}$ | **81.3**$_{\uparrow22.5}$ |

Table 2: **Comprehensive evaluation of baseline models and our proposed method on *ViDoRe*.** Results are presented using nDCG@5 metrics, and illustrate the impact of different components. Text-only metrics are not computed for benchmarks with only visual elements.

## 5.1 PERFORMANCE (R1)

We show performance is achieved iteratively through the combination of three factors; (1) a carefully crafted task-specific dataset, (2) pairing a pretrained LLM to a vision model to better leverage text semantics from the image, and (3) using multi-vector embeddings rather than a single vector representation to better capture the vast amount of visual information present in a document.

**Fine-tuning a Vision Model on a document retrieval oriented dataset: *BiSigLIP*.** SigLIP[9] is a strong vision-language bi-encoder producing single vector embeddings, and pretrained on billions of image-text pairs from the English split of WebLI (Chen et al., 2023). Further fine-tuning the textual component of this model on our document-oriented dataset (BiSigLIP) yields clear improvements across the board, particularly on figure retrieval (ArxivQA) and table retrieval tasks (TabFQuAD).

**Feeding image patches to a LLM: *BiPali*.** In the PaliGemma model architecture, SigLIP-generated patch embeddings are fed to a text language model and we can obtain LLM contextualized output patch embeddings.[10] This technique aligns the image token representations with the text token embeddings in the LLM's embeddings space, and augments the vision model embeddings with the language model's text understanding capabilities. We average pool these representations to obtain a single dense vector, effectively creating a PaliGemma bi-encoder model (BiPali). After fine-tuning on the training dataset, we obtain a model that performs slightly worse in English than the tuned BiSigLIP variant.[11] However, we see notable improvements in French tasks, indicating that BiPali's LLM (Gemma 2B) helps multilingual text understanding. This is particularly notable as our training dataset does not contain non-English samples.

**Leveraging Multi-Vector Embeddings through Late Interaction: *ColPali*.** One benefit of inputting image patch embeddings through a language model is that they are natively mapped to a latent space similar to the textual input (query). This enables leveraging the ColBERT strategy to construct one

---

[9]https://huggingface.co/google/siglip-so400m-patch14-384

[10]Note that the SigLIP model used in PaliGemma slightly differs in terms of number patches - 1024 patches for PaliGemma's vision encoder, and 729 for the standalone SigLIP model.

[11]This can be explained by the fact that contrary to SigLIP, the original PaliGemma is not trained on contrastive matching tasks, but rather on next token prediction. Our contrastive fine-tuning phase on 119K images to transform PaliGemma into a bi-encoder is 5 orders of magnitude smaller than SigLIP's original contrastive training.

embedding per image patch token, and at inference compute all interactions between text tokens and image patches, resulting in a step-change improvement in performance compared to BiPali. Results in Table 2 show that our *ColPali* model also largely outperforms the strong baselines based on `Unstructured` and captioning, as well as all evaluated text-image embedding models. The difference is particularly stark on the more visually complex benchmark tasks, such as InfographicVQA, ArxivQA, and TabFQuAD, respectively representing infographics, figures, and tables. However, text-centric documents are also better retrieved by the *ColPali* models across all evaluated domains and languages, making our approach the overall best-performing document-retrieval model.

**Negative Results.** For extensiveness, we also train ColSigLIP, a late interaction variant of the BiSigLIP model but obtain abysmal performances. We attribute this to the large gaps w.r.t. SigLIP's pre-training, in which only a pooled latent representation is used in the contrastive loss, which does not optimize the representations of individual patch and token embeddings. Similarly, we train a BiSigLIP$_{PaliGemma}$ variant, in which we retrieve the image representations from the SigLIP model that has been further updated by PaliGemma fine-tuning, and use the text representations from PaliGemma's text model. After fine-tuning on our dataset, performance is severely inferior to SigLIP$_{Vanilla}$ which simply encodes with SigLIP's original text and vision components. This indicates a logical misalignment between SigLIP embeddings, and Gemma embeddings after PaliGemma training. We detail these results in subsection C.1.

## 5.2 Latencies & Memory Footprint

**Online Querying. (R2)** Logically, querying latencies differ between *ColPali* and a BGE-M3 embedding model. For BGE, encoding takes about 22 ms for 15 tokens, while encoding a query with *ColPali*'s language model takes about 30 ms[12]. For smaller corpus sizes, computing the late interaction operation induces marginally small overheads ($\approx 1$ ms per 1000 pages in the corpus), and the cosine similarity computation between bi-encoder vectors is even faster. Optimized late interaction engines (Santhanam et al., 2022; Lee et al., 2023) enable to easily scale corpus sizes to millions of documents with reduced latency degradations.

**Offline Indexing. (R3)** Standard retrieval methods using bi-encoders represent each chunk as a single vector embedding, which is easy to store and fast to compute. However, processing a PDF to get the different chunks is the most time-consuming part (layout detection, OCR, chunking), and using captioning to handle multimodal data will only exacerbate this already lengthy process. On the other hand, *ColPali* directly encodes pages from their image representation. Although the model is larger than standard retrieval encoders, skipping the preprocessing allows large speedups at indexing[13] (Figure 2). As pages are embedded end-to-end in single forward pass, the VRAM usage depends exclusively on the sequence length (number of patches per image) which is fixed as well, enabling efficient batching strategies to fully leverage hardware acceleration. ColPali also benefits from most LLM efficiency improvements introduced in the ecosystem such as Flash Attention (Dao, 2023).

**Storage Footprint.** Our method requires storing a vector per image patch, along with 6 extra text tokens "Describe the image" concatenated to image patches. We project each PaliGemma vector to a lower dimensional space ($D = 128$) to maximize efficiency, leading to a memory footprint of 257.5 KB per page (subsection B.3). Importantly, the memory footprint of the naive ColBERT indexing strategy can be drastically improved through compression and clustering mechanisms (Santhanam et al., 2022; Clavié et al., 2024).

**Token pooling.** Token pooling (Clavié et al., 2024) is a CRUDE-compliant method (document addition/deletion-friendly) that aims amountto reduce the amount of multi-vector embeddings. For ColPali, many image patches share redundant information, e.g. white background patches. By pooling these patches together, we can reduce the amount of embeddings while retaining most information. Retrieval performance with hierarchical mean token pooling on image embeddings is shown in Figure 3 (left). With a pool factor of 3, the total number of vectors is reduced by 66.7% while 97.8% of the original performance is maintained. We note that the Shift dataset—composed of the most text-dense documents—is a clear outlier, showcasing more information dense documents contain less redundant patches and may be prone to worse performance degradation with such pooling techniques.

---

[12]Computed for a batch size of 1 (online), and averaged over 1000 queries. See subsection B.4

[13]Measures a NVIDIA L4 GPU, averaged on 100 pages, with a batch size of 4 pages for *ColPali* and 8 text chunks for Bi-Encoders. On average, a page is divided into 2.1 chunks. See subsection B.4.

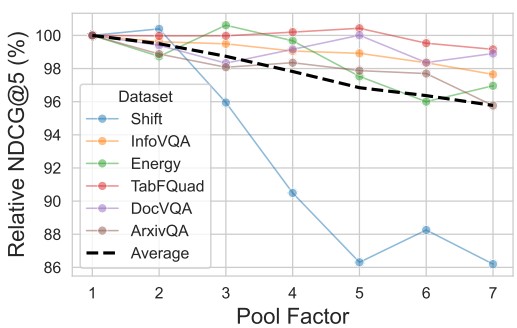 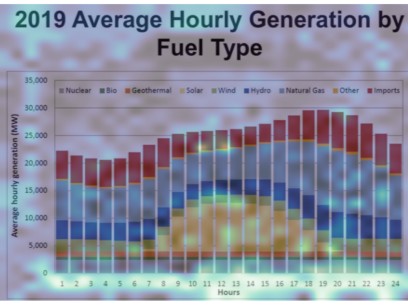

**Query:** "Which hour of the day had the highest overall eletricity generation in 2019?"

Figure 3: (Left: **Token Pooling**) Relative performance degradation when reducing the number of stored embeddings per document. (Right: **Interpretability**) For each term in a user query, *ColPali* identifies the most relevant document image patches (highlighted zones) and computes a query-to-page matching score.

## 5.3 INTERPRETABILITY

By superimposing the late interaction heatmap on top of the original image, we can visualize the most salient image patches with respect to each term of the query, yielding interpretable insights into model focus zones. As epitomized in Figure 3 (right), we observe *ColPali* exhibits strong OCR capabilities as both the words *"hourly"* and *"hours"* present a high similarity score with the query token <\_hour>. We also note particular focus on other non-trivial image features such as the x-axis representing hours being salient. Other visualization examples are shown in Appendix D.

## 6 ABLATION STUDY

We run various ablations to better understand the mechanisms at play. By default, result deltas reported below refer to nDCG@5 values averaged over all ViDoRe tasks. Detailed results in C.2.

**Tradeoffs between model size and the number of image patches.** We train a variant of PaliGemma with half the number of image patches (512). While we observe a clear performance degradation with respects to to the 1024-patch *ColPali* model ($-24.8$ nDCG@5), memory usage is much lower. As an alternative to PaliGemma, we train Idefics2-8B (Laurençon et al., 2024b), a VLM with a similar architecture and based on a Mistral-7B (Jiang et al., 2023) language backbone and a SigLIP vision encoder paired with a perceiver resampler. The most notable differences with PaliGemma lie in the size of the language model (2B and 7B resp.) and the number of image patches (between 512 and 2048 for PaliGemma, and 64 post-resampling for Idefics2[14]). Our results suggest better language models enable more efficient representations of image embeddings - ColIdefics2 with 64 patches largely outperforms out *ColPali* with 512 patches (+20.1 nDCG@5). However ColIdefics2 (64) remains less accurate than ColPali (1024) ($-4.7$ nDCG@5) while being about twice as slow in terms of training and inference latency. These results suggest there are *tradeoffs* between performance (R1), latencies during online querying (R2) and offline indexation phases (R3), and index memory size.

**Unfreezing the vision component.** We train a *ColPali* variant by also backpropagating through and updating the vision encoder and the projection layer. This leads to a slight performance degradation ($-0.7$ nDCG@5). These conclusions may change with larger scales of training data.

**Impact of "query augmentation" tokens.** In ColBERT, special tokens are concatenated to the input query to serve as soft query augmentation buffers. Training without these tokens, we observe no significant performance difference in the English benchmarks. However, performance on the French tasks seems to improve (+9.8 nDCG@G on Shift, +6.3 nDCG@5 on TabFQuAD, Table 7).

**Impact of the Pairwise CE loss.** Training with an in-batch negative contrastive loss, instead of the pairwise CE loss that only considers the hardest negative sample, leads to a slight performance degradation ($-1.6$ nDCG@5) on the aggregated benchmark.

---

[14]With the option of adding 4 sub-image crops of 64 tokens each to the sequence, for a total of 320 tokens

**Adapting models to new tasks.** Contrary to more complex multi-step retrieval pipelines, *ColPali* can be trained end-to-end, directly optimizing the downstream retrieval task which greatly facilitates fine-tuning to boost performance on specialized domains, multilingual retrieval, or specific visual elements the model struggles with. To demonstrate, we add 1552 samples representing French tables and associated queries to the training set. This represents the only French data in the training set, with all other examples being kept unchanged. We see clear nDCG@5 improvements ($+2.6$) and even starker Recall@1 gains ($+5$) on the TabFQuAD benchmark, with no performance degradation on the rest of the benchmark tasks ($+0.4$ nDCG@5 overall).

**Better VLMs lead to better visual retrievers.** As improved VLMs are released, it is interesting to observe if improved performances on generative tasks translate once these models are adapted for image retrieval tasks through ColPali training strategies. We train the recently released Qwen2-VL 2B (Wang et al., 2024b), a SOTA 2 billion parameter generative VLM, with the same data and training strategy, obtaining ColQwen2-VL. To approximately match ColPali's memory requirements, we limit the number of image patches to 768, slightly less than ColPali's 1024 patches. We observe clear performance improvements of $+5.3$ nDCG@5 values over ColPali showcasing clear performance correlations between generative benchmarks performance and retrieving metrics.

**Out-of-domain generalization.** Some of the datasets in the ViDoRe benchmark have train sets, which we have integrated within the ColPali train set (eg. academic tasks). This is standard in embedding models (Wang et al., 2024a; Lee et al., 2024), and while ColPali also exhibits strong performance on tasks in which this is not the case (French data is never seen by the model during training for instance), it remains interesting to evaluate model performance when training is done on a fully disjoint data distribution. We train a ColPali variant solely using the recent DocMatix dataset (Laurençon et al., 2024a), a large scale, synthetically annotated visual document question answering dataset, which we subsample to obtain a comparably-sized train set. Results on ViDoRe show the performance drop is minor ($-2.2$ nDCG@5), still outperforming the closest baseline method by over 12 points. These results showcase ColPali generalizes well outside of its training distribution, and demonstrate that our results are not unreasonably boosted with respect to baselines (BGE-M3) that cannot be fine-tuned on the same data[15]

## 7    CONCLUSIONS

In this work, we introduced the Visual Document Retrieval Benchmark (*ViDoRe*), which evaluates document retrieval systems in realistic settings involving visually complex documents. We demonstrated that current retrieval pipelines and contrastive vision-language models struggle to efficiently exploit visual information embedded in documents, leading to suboptimal performance. To address this, we presented *ColPali*, a novel retrieval method that leverages Vision-Language Models to create high-quality, multi-vector embeddings purely from visual document features. *ColPali* largely outperforms the best existing document retrieval methods while enabling faster corpus indexing times and maintaining low querying latencies, thus circumventing many pain points of modern document retrieval applications. We hope to drive industrial adoption, and to encourage future work by publicly releasing the *ViDoRe* benchmark, the data, the codebase, and all models and baselines from our work.

**Future Work.** Beyond performance improvements that could be obtained through better data, backbone models or training strategies, our vision at term is to combine visual retrieval systems and visually grounded query answering to create end-to-end RAG systems that purely function from image features. This idea is supported by concurrent work (Ma et al., 2024) showcasing the strong promises of VLMs for visual QA, and may eventually become a new industrial standard for document processing. In this line of work, reliability is key, and confidence estimation techniques for Information Retrieval methods could become central to implement abstention mechanisms (Gisserot-Boukhlef et al., 2024), and are particularly interesting given the information rich multi-vector scoring mechanisms of late interaction systems. Expanding benchmarking efforts to cover more languages, modalities, and tasks is also a crucial future research direction (Jiang et al., 2024).

---

[15]To train with data resembling the one BGE-M3 models would see at inference time would require running complex extraction pipelines for the more than 100K documents in the training set, notably relying on external proprietary captioning models which is both too costly and lengthy. This is not needed to train vision-based models.

## REPRODUCIBILITY STATEMENT

For transparency, reproducibility and to foster future work, we release our training data, model checkpoints (adapters), entire codebase, and complete evaluation benchmark under MIT licenses as detailed in the main paper. We also host a public *ViDoRe* leaderboard[16] to foster concurrent work in the space. The supplementary material further details training configurations for our models (also specified in HuggingFace model repositories), and dives into the process we used to generate synthetic data, how latency computations are performed, as well as provides further detailed evaluation results.

## ACKNOWLEDGEMENTS

This work is partially supported by Illuin Technology, and by a grant from ANRT France. This work was performed using HPC resources from the CINES ADASTRA through Grant 2024-AD011015443 and from IDRIS with grant 2024-AD011015724R1. We extend our warm thanks to Jonathan Dong, Caio Corro, Victor Pellegrain and Ender Konukoglu for their valuable feedback on the paper.

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

# A  BENCHMARK DATASETS

## A.1  ACADEMIC DATASETS

**DocVQA** (Mathew et al., 2020) includes collected images from the UCSF Industry Documents Library. Questions and answers were manually annotated.

**InfoVQA** (Mathew et al., 2021) includes infographics collected from the Internet using the search query "*infographics*". Questions and answers were manually annotated.

**TAT-DQA** (Zhu et al., 2022) is a large-scale Document VQA dataset that was constructed from publicly available real-world financial reports. It focuses on rich tabular and textual content requiring numerical reasoning. Questions and answers were manually annotated by human experts in finance.

**arXivQA** (Li et al., 2024) is a VQA dataset based on figures extracted from arXiv publications. The questions were generated synthetically using GPT-4 Vision.

**TabFQuAD** (Table French Question Answering Dataset) is designed to evaluate TableQA models in realistic industry settings. We create additional queries to augment the existing human-annotated ones using the same method described in subsection A.2.

## A.2  PRACTICAL DATASETS

**Methodology.** Creating a relevant retrieval dataset close to real use cases is a major challenge as the dataset needs to be both sufficiently large for effective fine-tuning and sufficiently diverse to cover a broad range of modalities (full text, tables, charts, ...), domains (industry, healthcare, ...), and query-document interactions (extractive questions, open-ended questions, ...). Our approach to building this dataset involves several steps: (1) we use a web crawler to collect publicly available documents on various themes and sources, (2) we convert these PDFs into a series of images, one per page, and (3) we generate queries related to each image using a VLM.

**Web-Crawler.** We implemented a web crawler to efficiently collect large volumes of documents related to a given topic. The crawler is seeded with a user-defined query (e.g. "artificial intelligence") and then uses GPT-3.5 Turbo to brainstorm related topics and subtopics. This query augmentation strategy aims at both broadening and deepening the search. GPT-3.5 Turbo is further used to generate diverse search queries from each subtopic. This query set is then consumed by a pool of parallel workers whose job is to fetch the associated most relevant documents. We use SerpAPI[17] along with a filetype filter (PDF documents only) to programmatically scrape Google Search rankings. Each file is hashed and stored in a Bloom filter (Bloom, 1970) shared among workers to avoid duplicate documents in the final corpus. Unique scraped files are downloaded, and inserted into a SQLite database along with additional metadata.

**Datamix.** Using the web crawler, we collected approximately 100 documents for each of the following four seeds: *"energy"*, *"government reports"*, *"healthcare industry"*, and *"artificial intelligence"*. These seeds were meticulously hand-picked to align with real-use cases for retrieval models and visually rich pages. We also removed all documents containing any private information.

**Query Generation.** To increase the efficiency of our query generation scheme and to limit API calls, we generate at most 3 questions per image. From all the documents collected, we randomly sample 10,000 images per theme and call Claude-3 Sonnet with the following prompt:

---

[17] `https://serpapi.com/`

```
You are an assistant specialized in Multimodal RAG tasks.

The task is the following: given an image from a pdf page, you will have to
generate questions that can be asked by a user to retrieve information from
a large documentary corpus.
The question should be relevant to the page, and should not be too specific
or too general. The question should be about the subject of the page, and
the answer needs to be found in the page.

Remember that the question is asked by a user to get some information from a
large documentary corpus that contains multimodal data. Generate a question
that could be asked by a user without knowing the existence and the content
of the corpus.
Generate as well the answer to the question, which should be found in the
page. And the format of the answer should be a list of words answering the
question.
Generate at most THREE pairs of questions and answers per page in a dictionary
with the following format, answer ONLY this dictionary NOTHING ELSE:

{
    "questions": [
        {
            "question": "XXXXXX",
            "answer": ["YYYYYY"]
        },
        {
            "question": "XXXXXX",
            "answer": ["YYYYYY"]
        },
        {
            "question": "XXXXXX",
            "answer": ["YYYYYY"]
        },
    ]
}
where XXXXXX is the question and ['YYYYYY'] is the corresponding list of answers
that could be as long as needed.

Note: If there are no questions to ask about the page, return an empty list.
Focus on making relevant questions concerning the page.
Here is the page:
```

**Human Validation.** We manually validate every single synthetically created query in *ViDoRe* to ensure quality, query relevance, and consistency with the benchmark objective of evaluating retrieval in practical industrial settings. During this step, we randomly assign document-pair queries to 4 volunteer annotators and instruct them to filter out queries that do not fit the above-listed criteria. We also instruct annotators to flag any documents they deem to contain PII information or content not suited for an academic benchmark. No flag was raised during the entirety of the process, validating our prior PDF collection strategy. 100 queries per topic are collected in this manner. Annotators are colleagues and collaborators of the authors who volunteered to help. Each annotator spent approximately 3 hours filtering the larger query set down to 100 high-quality queries per topic.

## A.3 TRAINING DATASET

The statistics of the train set are given in the following table. The creation of the train set follows the same methodology as in subsection A.2. We made sure that a PDF document cannot have pages in both the training set and the test set to prevent data leakage and that there are no duplicate documents in each split.

| Dataset Split | Split Size | Language | Domain |
|---|---|---|---|
| DocVQA | 39,463 | English | Scanned documents from UCSF Industry |
| InfoVQA | 10,074 | English | Infographics scrapped from the web |
| TATDQA | 13,251 | English | High-quality financial reports |
| arXivQA | 10,000 | English | Scientific Scientific Figures from arXiv |
| Scrapped PDFs | 45,940 | English | Varied PDFs from 3885 distinct URL domains |
| **TOTAL** | **118,695** | **English-only** | **Mixed** |

Table 3: Details on the different splits in the dataset used to train ColPali.

# B  IMPLEMENTATION DETAILS

## B.1  CODEBASE

The codebase is written in PyTorch[18] and leverages HuggingFace tooling for model implementations and trainers[19].

## B.2  HYPERPARAMETERS

Hyperparameters are tuned on a validation split composed of $2\%$ of the training dataset. We find bi-encoder methods to be more sensible to learning rate variations than late interaction-based models and achieve the best performance for all models with a learning rate of $5e - 5$. We experiment with LoRA rank and $\alpha$ values and do not notice particular improvements past $r = \alpha = 32$. Per-device batch sizes are kept small due to long sequence lengths that complicate scaling past $b = 4$. We simulate larger batch sizes with multi-GPU training and train with a total batch size $b = 32$ with no accumulation, for 1 epoch on our training set.

## B.3  EMBEDDING SIZE

Minimizing storage footprint can be essential to industrial retrieval systems if databases contain millions of documents. With this criterion in view, we have compared the embedding sizes of the models in our study. As shown in Table 4, *ColPali*'s embedding size is an order of magnitude larger than BM25 and two orders of magnitude larger than BGE-M3. However, in practical scenarios, pooling multi-vector embeddings by centroid cluster, or quantizing embeddings to binary representations [20] can reduce storage costs by two orders of magnitude (Santhanam et al., 2022) with minimal performance hits, and make storage costs competitive with other systems.

| Model | Embedding size (KB) |
|---|---|
| BGE-M3 | 8.60 |
| BM25 (dense emb.) | 3.00 |
| BM25 (sparse emb.) | $1.56 \pm 0.51$ |
| *ColPali* (float16) | 257.5 |

Table 4: Comparison of the embedding sizes for the DocVQA test set from *ViDoRe* w.r.t. different retrieval models. The *mean ± std* size is given for the sparse embeddings. In general multiple vectors (2-5) per page are used for BGE-M3 and BM25.

## B.4  LATENCY COMPUTATIONS

To ensure comparison fairness, the latencies of the different retrieval systems shown in Figure 2 are measured on the same `g2-standard-8` GCP VM with a NVIDIA L4 GPU. Document pages are embedded using the highest settings of `Unstructured` with captioning (see subsection 3.2). SigLIP and *ColPali* are both loaded with `bfloat16` parameter dtypes. The reported times in Table 5 are the

---

[18] https://pytorch.org/

[19] https://huggingface.co

[20] https://blog.vespa.ai/scaling-colpali-to-billions/

average per-page latencies for each indexing operation on 1000 randomly chosen documents across all splits of the *ViDoRe* benchmark test set. A batch size of 8 was used for the BGE-M3 model used with `Unstructured`, and a batch size of 4 was used for SigLIP and *ColPali*.

| Indexing operation | Latency (s) | | |
|---|---|---|---|
| | **Unstructured** | **SigLIP** | *ColPali* |
| Layout detection | 0.81 | NA | NA |
| OCR | 2.67 | NA | NA |
| Captioning | 3.71 | NA | NA |
| Page encoding | 0.03 | 0.12 | 0.39 |
| Total | 7.22 | 0.12 | 0.39 |

Table 5: Page-level latencies for document indexing using various retrieval systems. SigLIP and *ColPali* are much faster than `Unstructured` because they don't require the layout detection, OCR, and captioning operations.

## B.5 CAPTIONING

Examples of captions generated for visually rich document chunks with Claude-3 Sonnet are shown in Figure 5 and Figure 4. The prompt used for generating the description is the following:

> You are an assistant specialized in document analysis. Given a table or a figure, you have to provide a detailed summary of the content in maximum 3000 characters. Your summary should be qualitative and not quantitative. Here is the table/figure to analyze: {image}. Answer ONLY with the caption of the table/figure.

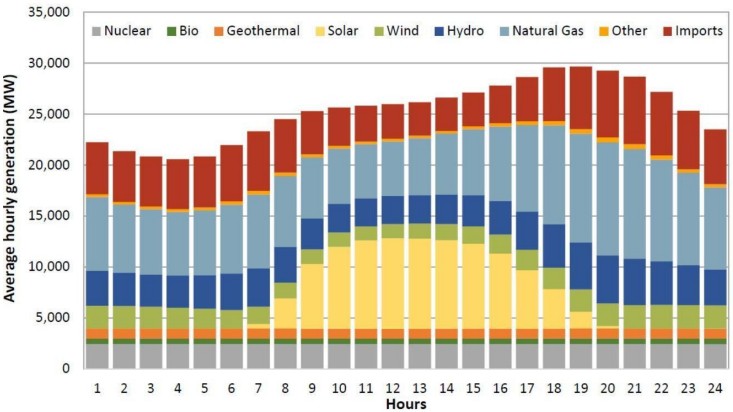

Figure 4: Example from the "Energy" test set.
**Caption:** *The image depicts the hourly energy generation profile, illustrating the contributions of various energy sources over 24 hours. The data is presented as a stacked bar chart, with the x-axis representing the hours of the day from 1 to 2, and the y-axis showing the average hourly generation in MW. The bars are segmented into different colors, each representing a distinct energy source: nuclear, bio, geothermal, solar, wind, hydro, natural gas, and other imports. The chart provides insights into the temporal variations in energy generation across different sources, highlighting the interplay between baseload and intermittent sources throughout the day.*

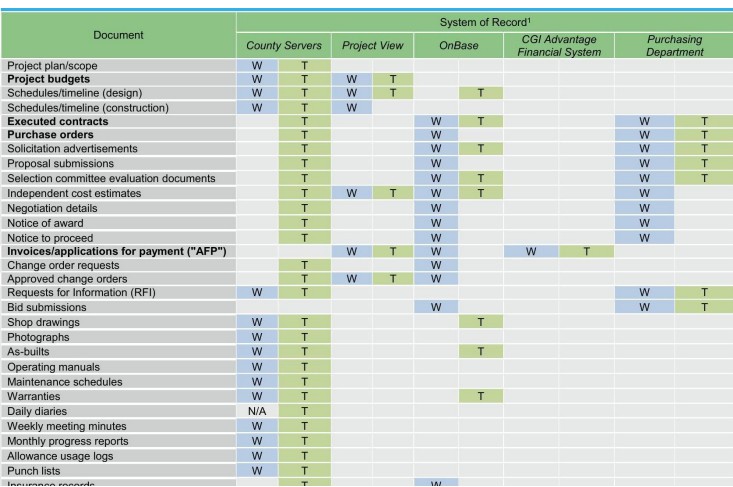

Figure 5: Example from the "Government Reports" test set.

**Caption:** *The image shows a table titled "System of Record" which outlines the different types of documents or records maintained across various systems or departments within an organization related to project management and construction. The rows list documents like project plans, budgets, schedules, contracts, purchase orders, invoices, change requests, bid submissions, drawings, manuals, meeting minutes, and reports. The columns indicate the system or department responsible for maintaining each record, such as County Servers, Project View, OnBase, CGI Advantage Financial System, and Purchasing Department. The table uses "W" and "T" markers to denote which system or department serves as the primary source (writer) or storage location (trailer) for each type of document.*

## C  ADDITIONAL RESULTS

### C.1  OTHER METRICS

| | ArxivQ | DocQ | InfoQ | TabF | TATQ | Shift | AI | Energy | Gov. | Health. | Avg. |
|---|---|---|---|---|---|---|---|---|---|---|---|
| **Unstructured** text-only | | | | | | | | | | | |
| BM25 | - | 26.6 | - | - | 34.6 | 45.0 | 86.0 | 70.0 | 68.0 | 74.0 | - |
| BGE-M3 | - | $22.8_{\downarrow3.8}$ | - | - | $26.1_{\downarrow8.5}$ | $51.0_{\uparrow6.0}$ | $81.0_{\downarrow5.0}$ | $72.0_{\uparrow2.0}$ | $67.0_{\downarrow1.0}$ | $77.0_{\uparrow3.0}$ | - |
| **Unstructured** + OCR | | | | | | | | | | | |
| BM25 | 26.7 | 28.9 | 54.0 | 30.4 | 50.0 | 52.0 | 86.0 | 77.0 | 74.0 | 80.0 | 55.9 |
| BGE-M3 | $28.1_{\uparrow1.4}$ | $22.9_{\downarrow6.0}$ | $53.8_{\downarrow0.2}$ | $55.7_{\uparrow25.3}$ | $38.6_{\downarrow11.4}$ | $\mathbf{56.0}_{\uparrow4.0}$ | $82.0_{\downarrow4.0}$ | $79.0_{\uparrow2.0}$ | $76.0_{\uparrow2.0}$ | $83.0_{\uparrow3.0}$ | $57.5_{\uparrow1.6}$ |
| **Unstructured** + Captioning | | | | | | | | | | | |
| BM25 | 35.5 | 30.2 | 61.5 | 24.3 | 49.0 | 47.0 | 79.0 | 76.0 | 75.0 | 81.0 | 55.9 |
| BGE-M3 | $29.3_{\downarrow6.2}$ | $26.0_{\downarrow4.2}$ | $62.1_{\uparrow0.6}$ | $58.6_{\uparrow34.3}$ | $30.6_{\downarrow18.4}$ | $55.0_{\uparrow8.0}$ | $80.0_{\uparrow1.0}$ | $78.0_{\uparrow2.0}$ | $69.0_{\downarrow6.0}$ | $83.0_{\uparrow2.0}$ | $57.2_{\uparrow1.3}$ |
| **Contrastive VLMs** | | | | | | | | | | | |
| Jina-CLIP | 19.4 | 7.3 | 26.7 | 12.5 | 1.6 | 2.0 | 11.0 | 13.0 | 15.0 | 17.0 | 12.6 |
| Nomic-vision | 10.4 | 6.7 | 22.1 | 9.6 | 1.6 | 0.0 | 9.0 | 9.0 | 7.0 | 13.0 | 8.8 |
| SigLIP (Vanilla) | 34.2 | 21.3 | 51.8 | 46.1 | 17.9 | 13.0 | 50.0 | 51.0 | 47.0 | 65.0 | 39.7 |
| **Ours** | | | | | | | | | | | |
| SigLIP (Vanilla) | 34.2 | 21.3 | 51.8 | 46.1 | 17.9 | 13.0 | 50.0 | 51.0 | 47.0 | 65.0 | 39.7 |
| BiSigLIP (+fine-tuning) | $49.2_{\uparrow15.0}$ | $23.8_{\uparrow2.5}$ | $59.0_{\uparrow7.2}$ | $52.1_{\uparrow6.0}$ | $20.7_{\uparrow2.8}$ | $16.0_{\uparrow3.0}$ | $62.0_{\uparrow12.0}$ | $61.0_{\uparrow10.0}$ | $55.0_{\uparrow8.0}$ | $72.0_{\uparrow7.0}$ | $47.1_{\uparrow7.4}$ |
| BiPali (+LLM) | $46.4_{\downarrow2.8}$ | $20.0_{\downarrow3.8}$ | $54.6_{\downarrow4.4}$ | $63.2_{\uparrow11.1}$ | $20.4_{\downarrow0.4}$ | $34.0_{\uparrow18.0}$ | $59.0_{\downarrow3.0}$ | $45.0_{\downarrow16.0}$ | $57.0_{\uparrow2.0}$ | $56.0_{\downarrow16.0}$ | $45.6_{\downarrow1.5}$ |
| *ColPali* (+Late Inter.) | $\mathbf{72.4}_{\uparrow26.0}$ | $\mathbf{45.6}_{\uparrow25.6}$ | $\mathbf{74.6}_{\uparrow20.0}$ | $\mathbf{75.4}_{\uparrow12.1}$ | $\mathbf{53.1}_{\uparrow32.7}$ | $55.0_{\uparrow21.0}$ | $\mathbf{93.0}_{\uparrow34.0}$ | $\mathbf{85.0}_{\uparrow40.0}$ | $\mathbf{85.0}_{\uparrow28.0}$ | $\mathbf{88.0}_{\uparrow32.0}$ | $\mathbf{72.7}_{\uparrow27.1}$ |

Table 6: **Comprehensive evaluation of baseline models and our proposed method on *ViDoRe*.** Results are presented using Recall@1 metrics. Text-only metrics are not computed for benchmarks with only visual elements.

### C.2  MODEL VARIANTS

| | ArxivQ | DocQ | InfoQ | TabF | TATQ | Shift | AI | Energy | Gov. | Health. | Avg. |
|---|---|---|---|---|---|---|---|---|---|---|---|
| ColSigLIP (PaliGemma) | 3.1 | 3.0 | 5.1 | 6.2 | 2.5 | 1.0 | 3.4 | 3.4 | 2.3 | 2.2 | 3.2 |
| BiSigLIP (PaliGemma) | 18.5 | 14.6 | 33.4 | 39.5 | 16.1 | 5.2 | 27.6 | 32.6 | 36.6 | 35.7 | 26.0 |
| ColSigLIP (Original) | 2.6 | 2.2 | 2.3 | 5.7 | 1.8 | 1.0 | 2.6 | 4.1 | 1.4 | 1.5 | 2.5 |
| ColPali (No Q.A. Tokens) | 80.4 | 53.2 | 82.4 | 77.4 | 65.7 | 63.4 | 97.0 | 89.9 | 93.6 | 92.4 | 79.6 |
| ColPali (Docmatix) | 71.3 | 48.0 | 80.0 | 83.9 | 59.1 | 73.8 | 95.7 | 93.8 | 92.5 | 93.1 | 79.1 |
| ColPali (224) | 71.0 | 37.4 | 62.3 | 65.7 | 28.6 | 20.4 | 65.7 | 66.8 | 73.9 | 73.0 | 56.5 |
| ColPali (Vision Trained) | 78.8 | 53.9 | 81.3 | 81.7 | 64.4 | 70.6 | 95.3 | 91.7 | 93.5 | 94.7 | 80.6 |
| ColPali (No Pairwise) | 79.0 | 53.0 | 82.1 | 85.3 | 63.2 | 66.2 | 94.9 | 88.9 | 92.7 | 92.1 | 79.7 |
| ColPali (+TabFQuAD training) | 77.6 | 54.7 | 82.6 | 86.5 | 65.4 | 73.9 | 94.8 | 92.4 | **94.2** | 94.8 | 81.7 |
| ColIdefics2 (64) | 73.6 | 48.0 | 82.4 | 81.6 | 63.0 | 57.2 | 95.5 | 86.9 | 86.6 | 91.2 | 76.6 |
| ColQwen2 (768) | **86.4** | **56.2** | **89.8** | **88.7** | **75.2** | **85.7** | **98.8** | **94.8** | 93.6 | **97.3** | **86.6** |
| *ColPali* (Reference: 448) | 79.1 | 54.4 | 81.8 | 83.9 | 65.8 | 73.2 | 96.2 | 91.0 | 92.7 | 94.4 | 81.3 |

Table 7: **Benchmark scores for the "negative results" and various ablations on *ViDoRe*; *ColPali* for reference.** Results are presented using nDCG@5 metrics. Text-only metrics are not computed for benchmarks with only visual elements.

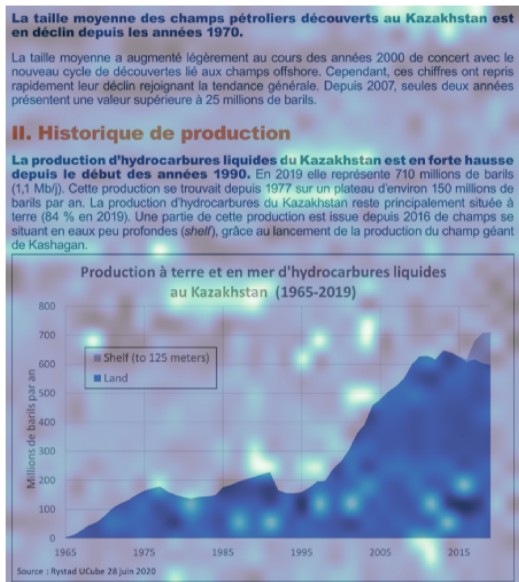

**Query:** "Quelle partie de la production pétrolière du Kazakhstan provient de champs en mer ?"

Figure 6: Similarity of the image patches w.r.t. the underlined token in the user query. This example is from the *Shift* test set.

## D   MORE SIMILARITY MAPS

In Figure 6, *ColPali* assigns a high similarity to all patches with the word *"Kazakhstan"* when given the token <_Kazakhstan>. Moreover, our model seems to exhibit world knowledge capabilities as the patch around the word *"Kashagan"*—an offshore oil field in Kazakhstan—also shows a high similarity score.

It is also interesting to highlight that both this similarity map and the one displayed in Figure 3 (right) showcase a few white patches with high similarity scores. This behavior might first seem surprising as the white patches should not carry a meaningful signal from the original images. We believe the vectors associated with these patches share a similar role with the ViT registers (Darcet et al., 2023), i.e. these patches were repurposed for internal computations and stored the global information from the whole image.

## E   MODEL GLOSSARY

### SIGLIP

SigLIP (Sigmoid Loss for Language Image Pre-Training) builds upon CLIP (Contrastive Language-Image Pretraining)—a foundational model that aligns images and text by maximizing the similarity between correct image-text pairs while minimizing it for incorrect ones, leveraging a contrastive loss (Zhai et al., 2023). Unlike CLIP (Radford et al., 2021), which applies the softmax function to the logits, SigLIP uses the sigmoid activation function. This innovation eliminates the need for a global view of all pairwise similarities between images and texts within a batch, enabling more flexible batch size scaling (up to 1M items per batch, with an effective optimal batch size of 32k). This approach allows SigLIP to achieve state-of-the-art performance in zero-shot image classification tasks.

### PALIGEMMA

PaliGemma is a 3B-parameter vision-language model. It integrates the SigLIP vision encoder with a Gemma-2B language decoder, connected via a multimodal linear projection layer (Lucas Beyer* et al.,

2024). The model processes images by segmenting them into a fixed number of Vision Transformer (Dosovitskiy et al., 2020) tokens, which are prepended to an optional text prompt.

A distinguishing feature of PaliGemma is its operation as a Prefix-Language Model (Prefix-LM). This design ensures full attention between image tokens and the user-provided input (prefix) while generating outputs auto-regressively (suffix). This architecture allows image tokens to access the task-specific query during processing, facilitating more effective task-dependent reasoning.

PaliGemma was trained in four stages: unimodal pretraining with existing components, extended multimodal pretraining, short high-resolution pretraining, and task-specific fine-tuning.

COLBERT

ColBERT (Contextualized Late Interaction over BERT) is a retrieval model designed to balance speed and effectiveness in information retrieval tasks (Khattab & Zaharia, 2020). Traditional retrieval models are typically categorized based on their type of interaction: either processing queries and documents independently for efficiency (bi-encoders) or jointly to capture rich contextual relationships (cross-encoders). ColBERT combines the advantages of both approaches through a novel late interaction mechanism.

Queries and documents are encoded separately using BERT, enabling offline pre-computation of document representations for scalability. Instead of pooling embeddings into a single vector, ColBERT retains token-level embeddings and employs a MaxSim operator to compute fine-grained similarity scores. For each query token, the model determines the maximum similarity with document tokens, summing these scores to compute relevance.

This architecture preserves the contextual richness of deep language models while significantly improving computational efficiency. By delaying the interaction step, ColBERT supports vector similarity indexing, facilitating end-to-end retrieval from large collections without prohibitive costs. Empirical evaluations on passage search datasets demonstrate that ColBERT achieves competitive effectiveness compared to existing BERT-based models (Devlin et al., 2018), while executing queries orders of magnitude faster and with drastically reduced computational requirements.

# F  EXAMPLES FROM THE *ViDoRe* BENCHMARK

## Energy

**Query**: What types of accounts or products allow investors to defer paying taxes?

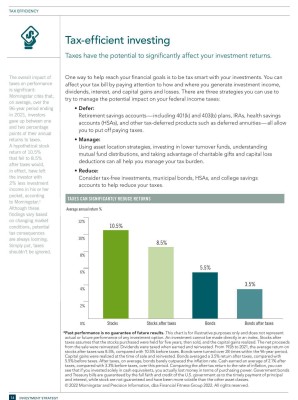

**Query**: What is the estimated total savings for a PV system in Durham under the net metering (flat rate) billing option over the system's useful life of 25 years?

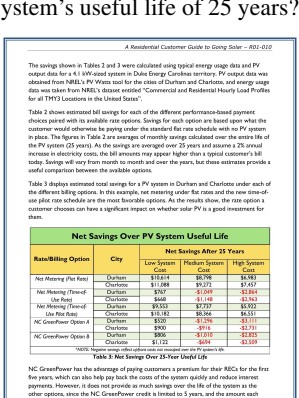

**Query**: What is the projected peak electricity demand in California for the year 2030?

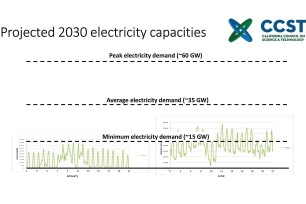

## Artificial Intelligence

**Query**: What are some common outcome areas targeted by TAII for different age groups?

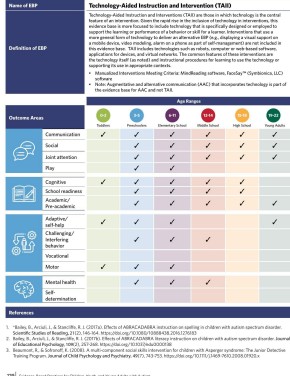

**Query**: What did the robot monitor to determine when to activate or deactivate the blower motor and blinker?

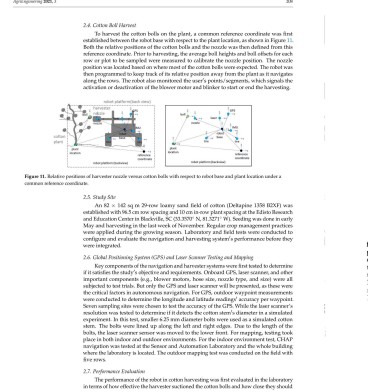

**Query**: What is the key approach used in the PDP architecture?

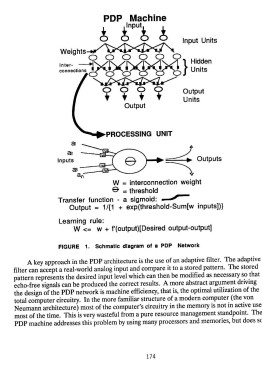

## Healthcare Industry

**Query**: What is the chemical formula for the ferroelectric material Lead Zirconium Titanate (PZT)?

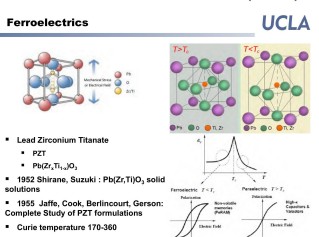

**Query**: What government entities are involved in public financing for healthcare in the US?

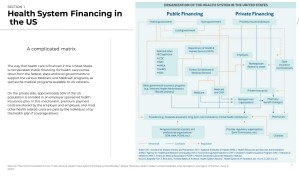

**Query**: What does the AVPU scale stand for in assessing the level of consciousness of a seriously ill child?

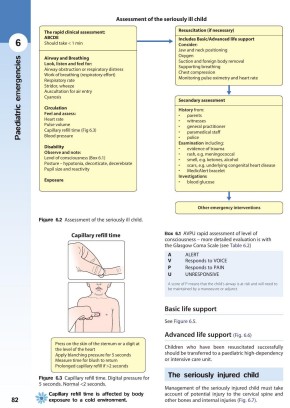

## Government Reports

**Query**: What are some mandates for the EPA under the Pollution Prevention Act?

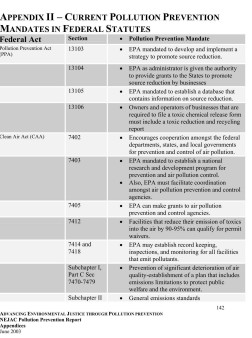

**Query**: What is the strategy of KPMG Hazem Hassan?

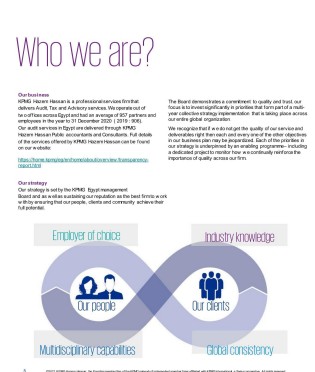

**Query**: What is the trust signal score for the consumer industry best-in-class archetype?

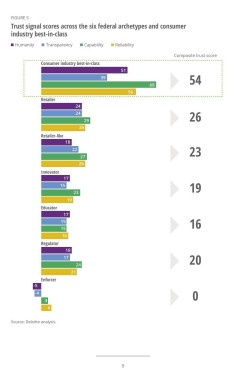

# Shift

**Query**: Selon le graphique, quelle est la capacité d'import et la consommation réelle de carburants SAF (biocarburants durables pour l'aviation) prévues en 2050 ?

**Query**: Quelle partie de la production pétrolière du Kazakhstan provient de champs en mer ?

**Query**: Quels sont les pays ayant la plus grande part des découvertes cumulées de pétrole brut en 2020 (en milliers de barils, hors découvertes cumulées) ?

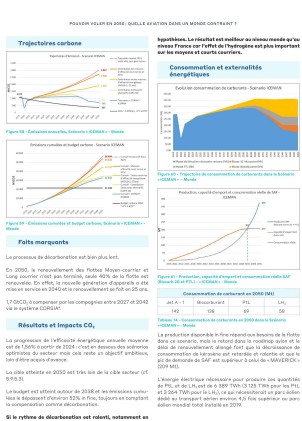

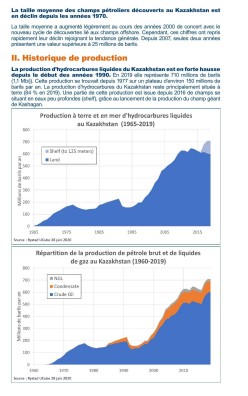

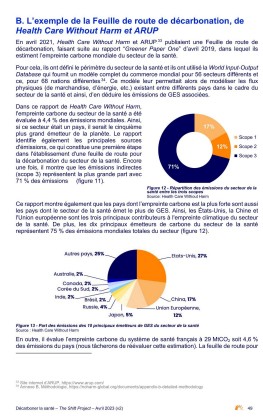

