# OpenReview forum: "ColPali: Efficient Document Retrieval with Vision Language Models"
_ICLR.cc/2025/Conference — ICLR 2025 Poster_

### Official Review · Reviewer_7YUV · 2024-10-29

**Soundness:** 3
**Presentation:** 2
**Contribution:** 2
**Rating:** 3
**Confidence:** 5

**Summary:**

The paper introduces ColPali, a document retrieval model that leverages LVLMs to create high-quality contextualized embeddings from images of document pages. This approach allows for efficient and fast query matching, leading to improved performance in document retrieval tasks. The paper's main contributions are: 1. The ViDoRe Benchmark: a comprehensive benchmark for evaluating document retrieval systems across various domains, visual elements, and languages. 2. The ColPali Model: a model that indexes documents based on their visual features and uses a late interaction mechanism for query matching, outperforming existing retrieval systems in terms of speed and accuracy.

**Strengths:**

**Performance Improvements**: ColPali demonstrates superior performance on the ViDoRe benchmark compared to modern document retrieval pipelines and is significantly faster, making it suitable for practical applications.

**End-to-End Trainability**: The ColPali model is end-to-end trainable, simplifying the optimization process for the retrieval task.

**Open-Source Release**: The authors release all project artifacts, including models and code, to encourage further development and research in the field.

**Multilingual Capabilities**: ColPali shows the ability to generalize to non-English languages, expanding its potential use cases globally.

**Weaknesses:**

**Benchmark Samples**: The manuscript should include examples from the ViDoRe benchmark that showcase its diversity in modalities, thematic domains, and language types. This will provide a more comprehensive understanding of the benchmark's scope and applicability.

**Retrieval Candidate Space Clarification**: The manuscript does not clearly define the retrieval candidate space for the ViDoRe benchmark. Clarifying this aspect is essential for understanding the benchmark's parameters and the retrieval process.

**ColPali Model Details**: The manuscript lacks essential details about the ColPali model. For instance, the definitions of N_q and N_d
  are not provided, nor is it explained how the document representation is generated. It is unclear whether the representation is derived from features corresponding to the [BOS] or [EOS] tokens, or if it is an average of the entire sequence features. These details are vital for replicating the study and understanding the model's workings.

**Broader Evaluation**: The evaluation is currently limited to the ViDoRe benchmark, with no experiments conducted on publicly available datasets such as those used for the retrieval task in VLM2Vec [1] and page prediction in MPdocVQA [2]. Expanding the evaluation to include these datasets would strengthen the experimental results and increase their convincing.


[1] Jiang Z, Meng R, Yang X, et al. VLM2Vec: Training Vision-Language Models for Massive Multimodal Embedding Tasks[J]. arXiv preprint arXiv:2410.05160, 2024.

[2] Tito R, Karatzas D, Valveny E. Hierarchical multimodal transformers for multipage docvqa[J]. Pattern Recognition, 2023, 144: 109834.

**Questions:**

**Page-Level Retrieval Capability**: Does the ColPali support page-level retrieval within a PDF, such as those in DUDE and MPdocvqa? Both MPdocvqa and DUDE also provide the gold document of a PDF.

**Text Retrieval Functionality**: Is ColPali capable of performing conventional text-based retrieval, or is its functionality limited to document images and visual features?

---

> ### Author Response · Authors · 2024-11-20
>
> We thank the reviewer for their feedback, which highlights our contributions in terms of benchmarking and methods along the concept of vision-based document retrieval first introduced in this work.
>
> **(W1) Benchmark Samples:** Figure 3 (right) and Figure 6 illustrate two varied query-page pairs present in ViDoRe, and an additional document example is given in Figure 5. Furthermore, the modalities, thematics, and languages of our benchmark are detailed in Table 1 and provide a comprehensive understanding of the benchmark's scope. Note that the ViDoRe benchmark can be easily visualized using the HuggingFace dataset viewer, which we will link to in the final manuscript. Encouraged by the reviewer, we have added additional query-page illustrations in the appendix.
>
> **(W2) Retrieval Candidate Space Clarification:** The retrieval candidate space for each ViDoRe subtask is explicitly defined in Table 1, detailing the language, number of queries, number of candidate documents and giving a brief modality and thematic overview of each of the subtasks.
>
> **(W3) ColPali Model Details:** As indicated throughout the paper, ColPali produces multiple vector embeddings for each page and for each queries. As indicated in the reviewer’s summary, we leverage the Late Interaction operation to match the (N_q) multiple query vectors to the (N_d) document vectors to produce a scalar matching score. This operation is detailed in Sec 4.1 L284, illustrated in Figure 1, and mentioned throughout the paper. We believe clearing up  this potential misunderstanding immediately explains all other remarks under this bullet point, as it should thus appear more clearly how document representations are generated (L280-283, Fig. 1, L378), to what correspond N_q and N_d (L284-288), and the fact neither <bos>, <eos>, nor pooling is necessary since all token embeddings are used. Pooling for the single-vector BiPali is detailed L363. We have clarified the notations (L286) and further re-explicited that all token embeddings are kept (L280). We note that our codebase is made fully available, and has since the paper’s release led to multiple successful replication attempts.
>
> **(W4) Broader Evaluation:** The ViDoRe benchmark is already quite exhaustive, and it does in fact already include multiple popular publicly available datasets (DocVQA, InfoVQA, TAT-DQA, ArxivQA) (L194), in addition to several other custom tasks spanning various languages and modalities. The reviewer suggests using datasets from the VLM2Vec paper but they are partially redundant with ViDoRe - some of the most relevant subtasks are already included in our benchmark (InfoVQA, DocVQA) -, and the preprint was released one week after the ICLR deadline. As for MP-DocVQA, this dataset is an extension of DocVQA in which a few additional context pages are added to the correct gold page. This dataset is originally designed for multi-page VQA, and since we have repurposed DocVQA for retrieval and already consider 500 pages as potential candidates for each query, we believe this task to be redundant, hence why it is not included in the current version of ViDoRe.
> We believe the 10 current ViDoRe subtasks that cover multiple languages (French, English), modalities (pure text, tables, figures, infographics, mixed-modality documents) and topics, are already a fairly diverse set of experiments and highly correlate with model performance. We acknowledge the VLM2Vec benchmarking efforts are complementary to the research direction introduced in this paper and have added a mention in the future work section.
>
>
> **Questions**
>
> **Page-Level Retrieval Capability:** As made clear thoughout the paper, yes, ColPali retrieves at a page-level granularity. DUDE is out of the scope of this work, as it is intended for multi-page QA and features multi-hop questions that necessitate multiple pages to be answered, as well as questions with no answers within the corpus. It is also largely similar to DocVQA which is already included in our benchmark. MP-DocVQA is mentioned in (W4).
>
> **Text Retrieval Functionality:** ColPali generates embeddings for both text (using only a LLM) and images (by feeding soft tokens produced by a vision model to the LLM) in a shared embedding space. As a result, beyond text-to-image retrieval, it is natively capable of performing text-to-text retrieval (similar to ColBERT), as well as image to image matching. Although we are aware of successful efforts in this direction, benchmarking this is beyond the scope of this paper which focuses on text-to-document image retrieval, which we believe has the potential to truly transform current document RAG practices This is being confirmed by the significant enthusiasm we have observed from both the industrial and academic community since the release of our preprint.
>
>
> We hope that under the light of our clarifications, the reviewer will reconsider their assessment of our paper and their score, and thank them once again for their review.

---

### Official Review · Reviewer_rAjC · 2024-11-04

**Soundness:** 3
**Presentation:** 3
**Contribution:** 3
**Rating:** 5
**Confidence:** 4

**Summary:**

In this paper, the authors explore a different approach to visually-rich document retrieval. To better assess current systems for this task, they construct a benchmark, called ViDoRe, which consists of various page-level retrieval tasks spanning multiple domains, languages, and practical settings. Inspired by ColBERT, the authors propose to perform document retrieval by directly embedding the images of the document pages. The proposed model, termed ColPali, is a Vision Language Model with a late interaction matching mechanism. The proposed model is evaluated on the ViDoRe benchmark and compared with existing pipelines for document retrieval.

**Strengths:**

1. This work investigates a very important problem: retrieving information from visually-rich documents. Visually-rich documents are quite common in practice and usually with complex layouts as well as informative elements such as figures and tables, which cannot be well handled by existing pipelines for document retrieval.
2. Different from previous pipelines, the authors accomplish visually-rich document retrieval by directly embedding the images of the document pages with VLMs and late interaction mechanisms. Experiments demonstrate the effectiveness and efficiency of the proposed ColPali model.
3. The proposed ViDoRe benchmark is a good contribution to the community as it can be used to evaluate systems on page-level document retrieval with a wide range of domains and applications, in the setting of simultaneous textual and visual understanding.
4. Overall, the paper is well-written. The main idea and key technical details are clearly presented.

**Weaknesses:**

1. The proposed pipeline is mainly inspired by ColBERT and PaliGemma. The author should give more explanations and analyses to prove the novelty of it.
2. The authors specially emphasize the importance of retrieval efficiency in industrial applications. However, the quantitative results regarding the latencies of different document retrieval model/systems are not detailed in the paper.

**Questions:**

The authors are encouraged to further explain the novelty of the proposed ColPali and give more quantitative results regarding the latencies of different document retrieval model/systems.

---

> ### Author Response · Authors · 2024-11-20
>
> We thank the reviewer for their comments that highlight the importance of our work and the effectiveness of our methods.
>
>
> **Novelty**
>
>  We believe our work to be a true paradigm shift for RAG systems. Since the release of our preprint, we have observed a huge interest in both the academic and industrial community in the “document retrieval from images” concept we introduce in this work. Although we do leverage pretrained models (PaliGemma) as building blocks, and existing multi-vector matching techniques (ColBERT), *our work is the first to consider constructing document embeddings from images with vision models, and to have showcased the feasibility and many benefits of this approach*. In the paper, we show this is enabled by several techniques we introduce and successfully apply to this document retrieval use case for the first time, namely
> - Repurposing generative vision-language models that align image token embeddings to the text token latent space and enable understanding text semantics “from image pixels”
> - Largely boosting performance by leveraging cross-modal multi-vector embeddings that better encode rich visual information
> - Introducing and releasing new synthetic datasets for this purpose and releasing all project artifacts
>
> Our released models enable huge boosts for many existing use cases–along both the retrieval performance and the indexing speed axes–, enable many novel future explorations, and have truly demonstrated a promising alternative to the previous document RAG standards prior to our preprint.
>
>
> **Latency Details**
>
>  The reviewer indicates the latency results are poorly detailed in the paper. In the submission, quantitative latency results for image embeddings are given in Section 3.2 and illustrated in Figure 2. As indicated in Figure 2’s caption, appendix Section B.4 further complements these results by offering expanded details on how these latencies were measured. To address the reviewer’s concerns, we have moved some details in the main paper and expanded the appendix with additional details. Query latencies are also detailed in section 5.2 (L399). Modifications are made in red.
>
>
> We thank the reviewer once again for their suggestions, and hope this rebuttal and the induced modifications have cleared some interrogations and improved the quality of our work. If this were to be the case, we kindly ask whether the reviewer would consider improving their review score.

---

### Official Review · Reviewer_nYTk · 2024-11-04

**Soundness:** 3
**Presentation:** 3
**Contribution:** 3
**Rating:** 8
**Confidence:** 2

**Summary:**

This paper introduces a novel document retrieval method called ColPali, which uses Vision Language Models (VLMs) to generate high-quality multi-vector embeddings directly from images of document pages. The method aims to address the performance bottlenecks of current text-centric retrieval systems when dealing with visually rich documents. The paper also introduces a new benchmarking framework, ViDoRe, to evaluate system performance in visually rich document retrieval.

**Strengths:**

1. The ViDoRe benchmark covers multiple domains and languages, providing a comprehensive framework to evaluate the capabilities of document retrieval systems.

2. ColPali presents an innovative concept that significantly improves performance through retrieval in the vision space. Experimental results demonstrate that ColPali significantly outperforms existing methods in several visually complex tasks and offers fast indexing and querying capabilities.

3. The paper mentions that all resources, including models, data, and code, are released under open licenses, which will promote further research and application within the community.

**Weaknesses:**

1. Although the ViDoRe benchmark covers multiple domains, the generality and adaptability of ColPali in broader application scenarios need further validation.

**Questions:**

No additional questions.

---

> ### Author Response · Authors · 2024-11-20
>
> We thank the reviewer for their rebuttal and appreciation of our work. In line with the reviewer's intuition, we can confirm that the open nature of our work and resources have been capital in promoting the development of our introduced concept of vision-based document retrieval since the release of our preprint.
>
> **Generality and Out-Of-Domain Application Scenarios.**
>
> We fully agree with the reviewer's assessment that validating the generalization capabilities of our method is of interest. In our paper, we have shown this in two ways:
>
> - **Out-of-distribution tasks**: Several tasks in the benchmark are by design excluded and out-of-distribution w.r.t. to the training set.  Typically, TabFQuAD and Shift Project are two tasks which are in the French language (all training data is in English), and covering modalities and topics not explicitly present in the training set (industrial tables, ecological topics). ColPali remains by far the best model on these domains, even though some baselines (Unstructured + BGE-M3) do cover French in their training set. This is a first step in showcasing the general nature of our technique.
>
> - **Extra Experiments**: To further confirm the insights detailed in (1) about the generalization capabilities of our method, we studied out-of-domain generalization (L502) through an extra experiment, in which we replace our (partially in-domain) training dataset by an entirely disjoint dataset comprised of synthetic data which is fully OOD w.r.t. the ViDoRe benchmark. As stated in the paper,  “results on ViDoRe show the performance drop is minor (−2.2 nDCG@5), still outperforming the closest baseline method by over 12 points. These results showcase ColPali generalizes well outside of its training distribution.”
>
> As benchmark results are often only part of the story, we have also released public demos hosted on HuggingFace spaces enabling users to test out ColPali for their given use cases and have had overwhelmingly positive feedback. Several preprints released since our preprint release have extensively and independently evaluated our ColPali method and have further demonstrated the very generalist nature of our model on previously unseen data distributions.
>
> We hope this rebuttal has reassured the reviewer regarding the perceived potential weaknesses of our work!

---

### Official Review · Reviewer_kCQZ · 2024-11-06

**Soundness:** 3
**Presentation:** 2
**Contribution:** 2
**Rating:** 5
**Confidence:** 4

**Summary:**

The paper presents two contributions in the context of document retrieval: (1) it presents ViDoRe, a new benchmark used to evaluate document retrieval algorithms with an emphasis on documents containing visual information, which is not covered by most of the existing benchmarks; and (2) it introduces ColPali, a model architecture that uses Vision-Language Models to efficiently index the documents in an efficient manner, and performing retrieval at query time with a competitive cost.

ViDoRe is composed from different existing academic datasets (mainly Vision Question Answering benchmarks), plus topic-specific publicly-accessible PDFs collected by the authors. ColPali is built based on PaliGemma-3B and the ColBERT strategy to generate a set of vision-text tokens from PDFs (during indexing) and text queries (during retrieval).

The results reported in the paper show that standard methods of document retrieval are either very expensive for indexing (e.g. solutions based on the off-the-self "Unstructured" tool, augmented with image captioning or OCR) or provide lower quality (e.g. approaches based on contrastive Vision-Language encoders). The proposed

**Strengths:**

- The proposed benchmark, ViDoRe, covers an important gap for evaluating existing document retrieval systems.
- The method presented in the paper significantly improves existing approaches, even those that require some fine-tuning (see table 2).
- The paper includes a section with extensive ablation studies that justify some of the decisions made during the design of the ColPali method.
- Hyperparameter tuning was not directly done on the ViDoRe benchmark, but on a 2% split from their training set (however, this training set poses some problems, see the weaknesses bellow).

**Weaknesses:**

- The paper builds ColPali iteratively starting from a SigLIP model, and using previously existing recipes. However, each of the steps is not very well detailed in the paper itself, which may make the paper hard to truly understand for readers not familiar with SigLIP, ColBERT, or PaliGemma. I would suggest expanding the details on section 5.1.
- For sytems that require any sort of tuning, the authors have used a dataset made of rouhgly 120k query-page pairs, 63% of which come from the same distribution used in the academic ViDoRe benchmark. The authors made sure that no query or page in the evaluation set is part of the training data, however the training data is very "in-distribution" of the evaluation data, which may bias both the hyperparameter tuning and the quality of the methods on ViDoRe. I would strongly suggest that the authors (also) present results, for the different methods that require some sort of tuning, using only the synthetic training data (including hyperparameter search). This way, we can measure if the gap between the proposed method and the Unstructured approaches is mainly due to the architecture or due to the (in-domain) fine-tuning  and hyperparameter selection.

**Questions:**

See questions / concerns implied in the weaknesses section.

---

> ### Author Response · Authors · 2024-11-20
>
> We thank the reviewer for their detailed review emphasizing the interest of our method and the value of our benchmark.
>
> **Clarifying the construction process**
>
> This work is the first to consider doing document retrieval purely from images, rather than extracted text content, marking a significant paradigm shift from previous techniques. Through the construction ablations in Section 5.1, we show that this breakthrough is possible through the combination of (1) a carefully crafted dataset that includes synthetic data, (2) pairing a pre-trained language model to the vision model to be able to capture text semantics from the image, and (3) using multi-vector embeddings rather than a single vector to better capture the vast amount of information present in a visual document.  To emphasize this point, we have added an introductory sentence to Section 5.1 and slightly clarified the structure, adding some details for readers less aware of the technical details of the underlying models. We have also added a model glossary in Appendix E further detailing SigLIP, PaliGemma and ColBERT for interested readers. Note that replication is also (and has been) easily enabled by the availability of the training code.
>
> **In-Distribution Training**
>
> The reviewer correctly notes that some tasks of the ViDoRe benchmark are in-distribution with respect to the training set. While this is common practice in embedding papers (MTEB evaluation), we were also interested in estimating the impact of in-domain training w.r.t. methodological innovations. This is done in three ways:
>
> - **Out-of-distribution tasks**: Several tasks in the benchmark are by design excluded and out-of-distribution w.r.t. to  the training set.  Typically, TabFQuAD and Shift Project are two tasks which are in the French language (all training data is in English), and covering modalities and topics not explicitly present in the training set (industrial tables, ecological topics). ColPali remains by far the best model on these domains, even though some baselines (Unstructured + BGE-M3) do cover French in their training set. This is a first step in showcasing the general nature of our technique.
>
> - **Finetuned baselines**: While being unable to finetune Unstructured-based methods because of the prohibitive cost of extracting text for the 40000+ pages of the training set with the same settings as what we run at inference (using layout detection, etc…), we do finetune the best vision-based baseline, the SigLIP model, on the same training data. This results in the BiSigLIP model, which presents performances that are vastly inferior to ColPali across the board (-22.7 nDCG@5). This shows the interest of our method goes beyond a strong dataset.
>
> - **Extra Experiments**: To further confirm the insights detailed in (1) and (2) about the generalization capabilities of our method, we studied out-of-domain generalization (L502) through an extra experiment, in which we replace our (partially in-domain) training dataset by an entirely disjoint dataset comprised of synthetic data (DocMatix) which is fully OOD w.r.t. the ViDoRe benchmark. As stated in the paper,  “results on ViDoRe show the performance drop is minor (−2.2 nDCG@5), still outperforming the closest baseline method by over 12 points. These results showcase ColPali generalizes well outside of its training distribution and demonstrate that our results are not unreasonably boosted with respect to baselines (BGE-M3) that cannot be finetuned on the same data.”
>
>
> We are thus very confident that the results reported are not due to in-domain training and that our method is very generalizable to OOD data. Furthermore, as we have released our training data, future work can leverage it (and already have) to disambiguate methodological innovations from data innovations.
>
>
> We hope this rebuttal has reassured the reviewer regarding the perceived weaknesses of our work, and hopefully incited them to consider improving their grade.

---

> > ### Comment · Reviewer_kCQZ · 2024-11-26
> >
> > **Clarifying the construction process**
> >
> > I appreciate very much the additional clarification provided in the updated manuscript. However, I still think that some significant details are left out of the paper / appendices.
> >
> > For instance, regarding BiPali. The updated manuscript states:
> >
> > > This technique aligns the image token representations with the text token embeddings in the LLM’s embeddings space, and augments the vision model embeddings with the language model’s text understanding capabilities.
> >
> > How _exactly_ is this aligment done? Can you refer to the section in the paper that explains it or relevant work that explain exactly how this is done?
> >
> > **In-Distribution Training**
> > Your arguments, in particular the last one, convinced me that the method is very likely generalizable to OOD data.
> >
> > If you address my first concern, which I think is crucial for reproducibility and building on top of the work, I'll gladly increase my score.

---

> > > ### Author Response · Authors · 2024-11-27
> > >
> > > We thank the reviewer for their response, and further attempt to address the remaining question in detail.
> > >
> > > > How exactly is this aligment done?
> > >
> > > In classic *contrastive vision-language models* ([CLIP](https://arxiv.org/abs/2103.00020), [SigLIP](https://arxiv.org/abs/2303.15343)), image and text representations are obtained through two different models, respectively an image encoder and a text encoder. Each model usually averages the multiple vectors they output (one vector per image patch, or respectively one per text token) to obtain a single “pooled” vector per modality. The obtained image representations and their corresponding text representation are aligned through a contrastive training objective. This is explained in the related work section (L135) in the paper, as well as in appendix section E (SigLIP).
> > >
> > > In modern *“generative” vision language models*, it is a bit different. Images are first fed through a pretrained vision encoder (e.g. a pretrained SigLIP), then the output vectors (one per image patch) are projected to a dimension corresponding to the text token embedding dimension of a given LLM. It thus becomes possible to feed “image tokens” to a LLM as if they were text tokens by concatenating the projected image patch embeddings with the text token embeddings and letting the LLM jointly process them. We briefly explain it in section 2.2 between lines 142 and 151, and represent it visually in Figure 1.
> > >
> > > In order to teach the LLM to understand and leverage this visual information not seen during text pretraining, an extra training phase is needed (modality alignment). This is done by providing the model an image and a text prompt (e.g.: “Describe the image”) that are concatenated as explained above, then training the LLM with the classic causal language modeling objective on the textual ground truth (here an image description). This process is now commonplace in modern VLMs, and training on document-related tasks has been shown to enable strong OCR and document understanding capabilities. Further details are given in the [PaliGemma paper](https://arxiv.org/abs/2407.07726) , but also in [LLaVA](https://arxiv.org/abs/2304.08485) and [Flamingo](https://arxiv.org/abs/2204.14198), all related work cited in the paper.
> > >
> > > In this work, our intuition was to exploit the representation of image patches *after* they have been processed by the language model (last hidden state of the LLM at each image token position). Thanks to the prior modality alignment training (done during PaliGemma training), the LLM can be used to “enrich” the image encoder's representations (L361). This is made possible by the LLM’s understanding of language and semantics acquired during its text pretraining phase. We show that through the contrastive finetuning described in section 4.1 (L293), we can induce the embedding of text tokens to align with the image patches that contain corresponding concepts. This can be seen by our interpretability experiments (Figure 3 right, L451), in which the embedding of the token “hour” is similar to the embedding of the image patch where the word “hour” is written. Showing this is possible, and then leveraging it for multimodal retrieval is a contribution of this work.
> > >
> > > We hope the reviewer deems these information to be sufficient. If this is the case, we will extend the glossary to expand on these points, notably with a more detailed description of the modality alignment training phase, and add a reference in the main paper.
> > > We thank the reviewer once again for their precious feedback.

---

### Author Response · Authors · 2024-12-03
**Summary of Review Process**

We briefly summarize the reviews and rebuttals to facilitate the AC’s assessment of our work.

**Strenghts.** All reviewers agree that our paper - which first introduces the concept of producing document embeddings from page images using Vision Language Models - tackles a “*very important problem*”, and the reviews highlight both the efficiency and performance gains of our proposed method ColPali, the value of our benchmark which “*fills an important gap*”, and appreciate our thorough resource release.

**Interrogations.** Reviewers kCQZ and nYTk had questions regarding the generalization of our method outside the training data distribution, which we thoroughly answered, notably pointing to an experiment in the paper designed to evaluate this. Reviewer kCQZ was satisfied by the response and expressed that they were convinced our work was “*very likely generalizable to OOD data*”.

**Improvements.** Reviewer rAjC acknowledges the paper is "*well written*" and "*clearly presented*", and we have further updated the manuscript to mainly:

-  Include additional background information on Vision Language Models and add minor clarifications to our construction process (kCQZ, rAjC)
- Move some latency details from the appendix to the main paper (rAjC)
- Add further dataset examples to the appendix, and mention related work released after the ICLR deadline (7YUV)

We have done our best to address all comments in detailed individual rebuttals, hopefully strengthening our work.  We thank all reviewers once again for their constructive comments and overall appreciation of our contribution.

---

### Meta-Review · Area_Chair_ZYNx · 2024-12-22

**Metareview:**

The paper has two main contributions toward document retrieval: the construction of ViDoRe benchmark and the VLM-based model called ColPali, both of which move away from the text-centric document retrieval paradigm. The reviewers find the benchmark significant and the experimental results convincing. Overall, the authors address concerns raised by the reviewers during the discussion period (see below). Both the concept introduced in this paper and the released models, data, code and benchmarks would be beneficial to the community.

**Additional Comments On Reviewer Discussion:**

Reviewer 7YUV gave the paper a score of 3; however, their concerns seem rather about details of the paper which are mostly already addressed. Reviewer kCQZ (score 5) and Reviewer nYTk (score 8)  have a concern about OOD generalization, which the authors addressed via additional experiments. Reviewer kCQZ (score 5) has another concern about the description of BiPaLI and alignment and promises to increase the score once addressed. Reviewer rAjC has a concern about novelty and empirical evaluation on latency which seems sufficiently addressed.

---

### Decision · Program_Chairs · 2025-01-22

Accept (Poster)